# A Simple, Rapid, and Effective Heparinase Protocol to Enable Nucleic Acid Study from Frozen Heparinized Plasma

**DOI:** 10.3390/mps6060112

**Published:** 2023-11-20

**Authors:** Rownock Afruza, Nicole Minerva, Justin B. Lack, Moumita Chakraborty, James A. Haddad, Rabab O. Ali, Christopher Koh, Elliot B. Levy, Ohad Etzion, Theo Heller

**Affiliations:** 1Translational Hepatology Section, National Institute of Diabetes and Digestive and Kidney Diseases, National Institutes of Health, Bethesda, MD 20892, USA; nicoleminerva757@gmail.com (N.M.); moumita.chakraborty@nih.gov (M.C.); jamesahaddad@gmail.com (J.A.H.); ohadet34@yahoo.com (O.E.); 2Research Technologies Development Section, National Institute of Allergy and Infectious Diseases, National Institutes of Health, Bethesda, MD 20892, USA; justin.lack@nih.gov; 3Liver Diseases Branch, National Institute of Diabetes and Digestive and Kidney Diseases, National Institutes of Health, Bethesda, MD 20892, USA; christopher.koh@nih.gov; 4Center for Interventional Oncology, Radiology and Imaging Sciences, Clinical Center, National Institutes of Health, Bethesda, MD 20892, USA; levyeb@cc.nih.gov

**Keywords:** cfRNAs, cell-free RNAs, qPCR, quantitative polymerase chain reaction, RT-qPCR, reverse transcription–quantitative polymerase chain reaction, NGS, next-generation sequencing, HCV, hepatitis C virus, cDNA, complementary DNA

## Abstract

Cell-free RNAs (cfRNAs) are promising analytes as non-invasive biomarkers and have even greater potential if tied in with metabolomics. Plasma is an optimal source for cfRNAs but is often derived from a variety of anticoagulants. Plasma obtained in heparin is suitable for metabolomics but is difficult to utilize for qPCR-based downstream analysis. In the present study, we aimed to develop a simple, time-efficient, and cost-effective heparinase protocol, followed by library preparation and sequencing of human plasma cfRNAs drawn and stored in heparin at −80 °C for several years. Blood was collected in CPT™ sodium heparin tubes from patients with chronic HCV infection (NCT02400216) at the National Institutes of Health (NIH) Clinical Center. Plasma cfRNAs were treated with heparinase I and used for library preparation and next-generation sequencing (NGS). Heparinase treatment maintained RNA integrity and allowed for successful library preparation for all the study subjects even with 7 ng of cfRNAs as starting material. The classification report derived from Pavian R package v1.2.0 showed no artificial reads. The abundance of chordate over microbial reads suggests no addition of experimental error through heparinase I treatment. We report a novel and practical approach to heparinase treatment for human plasma collected and frozen in sodium heparin for several years. This is an effective demonstration of utilizing heparin plasma for NGS and downstream transcriptomic research, which could then be integrated with metabolomics from the same samples, maximizing efficiency and minimizing blood draws.

## 1. Introduction

Advancements in technology have led to a revolution in metabolomics, transcriptomics, and proteomics, all now possible utilizing small amounts of plasma. A significant challenge is the choice of anticoagulant to derive the plasma, with ethylenediaminetetraacetic acid (EDTA), heparin, fluorides, and anticoagulant citrate dextrose (ACD) all widely available. Plasma drawn and stored in heparin is most similar to serum when utilized for metabolomics as compared to other anticoagulants such as ACD, EDTA, and fluorides [1]. On the other hand, heparin interferes with reverse transcriptase and restricts downstream analysis, such as qPCR and sequencing [2,3]. 

Cell-free RNAs (cfRNAs) are increasingly promising analytes in both cancer- and non-cancer-related research [4,5,6,7]. Successful removal of heparin and subsequent qPCR have been highly challenging, limiting use for the study of cfRNAs. Though heparinase enzymes can degrade heparin and thus allow downstream analysis, RNA often becomes denatured during incubation. RNAse inhibitors, MgCl_2,_ and different buffers could potentially prevent the degradation of cfRNAs. Successful heparinase treatment along with qRT-PCR and microRNA profiling in different body fluids and disease conditions has been previously demonstrated [3,8,9,10]. However, to the best of our knowledge, there has not yet been a study that has examined heparinase treatment on cell-free plasma RNA followed by library preparation and next-generation sequencing (NGS) in humans.

In the present study, we aimed to develop a simple, time-efficient, and cost-effective heparinase protocol, followed by library preparation and sequencing of plasma drawn and stored in heparin at −80 °C for several years from patients with chronic hepatitis C virus (HCV) infection.

## 2. Experimental Design

We have used human plasma from patients infected with HCV for different experiments in this study. Patients with chronic HCV infection were assessed and enrolled (NCT02400216) at the National Institutes of Health (NIH) Clinical Center [11]. The study was approved by The National Institute of Diabetes, Digestive and Kidney Diseases, the National Institute of Arthritis and Musculoskeletal Diseases, and the Institutional Review Board at NIH.

### 2.1. Materials

#### Human Plasma

Seven to eight mL of peripheral blood samples was obtained from an antecubital vein and drawn into CPT™ sodium heparin tubes (BD Biosciences, Ann Arbor, MI, USA, Cat# 362753) and spun down at 1600 rpm for 10 min at 4 °C for plasma. Then, 1 mL of plasma was aliquoted in multiple Eppendorf tubes, and all the samples were stored at −80 °C until analysis was complete. Plasma samples collected in sodium EDTA as well as in sodium heparin from anonymized healthy donors were obtained from the NIH Clinical Center Blood Bank.

## 3. Procedure

### 3.1. Extraction of RNA

cfRNAs were extracted from 1 mL plasma using a previously published method [7]. Per mL frozen plasma samples were mixed with 2 mL of RNAzol BD (MRC, Cat# RB192, Cincinnati, OH, USA) and 20 μL of Acetic acid (Sigma-Aldrich, Cat# A6283, St. Louis, MO, USA) in 5 mL DNA LoBind^®^ Tubes (Eppendorf, Cat# 0030122348, Hamburg, Germany), shaken vigorously for 30 s and incubated for 15 min. After that, 150 μL of 1–bromo–3–chloropropane (MRC, Cat# BP 151, Cincinnati, OH, USA) was added to the tubes, shaken vigorously for 30 s, and re-incubated for another 15 min in RT. Then, the mixture was centrifuged at 12,000× *g* (Centrifuge 5427 R, Rotor FA-45-12-17) for 15 min at room temperature, and the upper phase was placed in new 5 mL DNA LoBind^®^ Tubes for 2-propanol (Sigma-Aldrich, Cat# I9516-500ML, St. Louis, MO, USA) precipitation. Next, 1 μL of UltraPure™ Glycogen (Thermo Fisher Scientific, Cat# 10814010, Carlsbad, CA, USA) and 4 μL of UltraPure™ DNase/RNase-Free Distilled Water (Thermo Fisher Scientific, Cat# 10977023-10 × 500 mL, Grand Island, NY, USA) were added into the upper phase and 2-propanol mixture (1:1) to increase the recovery of cfRNAs during 2-propanol precipitation. After washing the pellet with 75% ethanol (Sigma-Aldrich, Cat# E7023-500ML, St. Louis, MO, USA) 4 times, it was dried in room temperature for 5 min and suspended in UltraPure™ DNase/RNase-Free Distilled Water. The concentration of extracted cfRNAs was measured using Quant-iT™ RNA Assay Kits and Quant-iT RNA HS Reagent (Cat# Q33225, Eugene, OR, USA) in a TBS-380 Mini-Fluorometer (Turner BioSystems, Sunnyvale, CA, USA).

#### 3.1.1. Treatment with Heparinase I Enzyme

cfRNAs suspended in UltraPure™ DNase/RNase-Free distilled water were incubated with *Bacteroides* heparinase I (NEB, Cat# P0735L, Ipswich, MA, USA) enzyme, 10X heparinase buffer (supplied with the *Bacteroides* heparinase I enzyme, NEB, Cat# P0735L, Ipswich, MA, USA), RNAse inhibitor (Applied biosystem, Cat# N8080119, Carlsbad, CA, USA), and NEBNext First Strand Synthesis Reaction Buffer (NEB, Cat# E7770L, Ipswich, MA, USA) for 1 h at 37 °C in a thermomixer (Eppendorf™ Thermomixer™ R, 9.75 L × 8.75 W × 4.75 H (25 cm × 22 cm × 12 cm)). Detailed concentrations and volumes added are displayed in Table 1.

#### 3.1.2. cDNA Synthesis, Library Preparation, and Sequencing

After 1 h of incubation at 37 °C, samples treated with *Bacteroides* heparinase I were taken for cDNA synthesis, followed by library preparation according to the manufacturer’s recommendations (NEB, Cat# E7770, Ipswich, MA, USA). Then, 4 μL of *Bacteroides* heparinase-I-treated sample was mixed with 4.6 μL of NEBNext First Strand Synthesis Reaction Buffer (5X) and 1.15 μL of NEBNext Random Primers followed by 15 min incubation at 94 °C in a pre-heated thermal cycler (BIO-RAD, Hercules, CA, USA, C1000 Touch Thermal Cycler) and then immediately placed on ice. After that, 2 μL of NEBNext First Strand Synthesis Enzyme Mix and 8 μL of nuclease-free water (supplied with the kit, NEB, Cat# E7770, Ipswich, MA, USA) were added to each sample and placed on the pre-heated thermal cycler for following cycles: 25 °C for 10 min, 42 °C for 50 min, 70 °C for 15 min, and 4 °C hold for 10 min. For second-strand cDNA synthesis, a master mix was made using NEBNext Second Strand Synthesis Reaction Buffer (10X) (8 μL), NEBNext Second Strand Synthesis Enzyme Mix (4 μL), and nuclease-free water (48 μL) for each sample. After adding the second-strand master mix to the sample, the mixture was well mixed with a pipette and incubated in the thermal cycler at 16 °C for 1 h. Then, the cDNAs were purified using SPRIselect beads (Beckman Coulter; Cat# B23317, Brea, CA, USA) and used for library preparation. Each library was mixed with 25 μL of NEBNext Ultra II Q5 Master Mix (provided with the kit, NEB, Cat# E7770, Ipswich, MA, USA) and was tagged using NEBNext Multiplex Oligos for Illumina (NEB, Cat# E6609) for PCR (Table 2) in the pre-heated thermal cycler. 

All the RNA libraries were checked using a High Sensitivity D5000 ScreenTape^®^ kit (Agilent, Cat#5067-5592, Santa Clara, CA, USA) in TapeStation (Agilent 4200 TapeStation System) for quality purposes and quantified using Quant-iT™ PicoGreen™ dsDNA Assay Kits (Thermo Fisher Scientific, Cat# P7581, Eugene, OR, USA) in a TBS-380 Mini-Fluorometer (Turner BioSystems, Sunnyvale, CA, USA). Libraries were sequenced on a NovaSeq6000 System (Illumina, San Diego, CA, USA) for 50 base pair (bp) paired-end reads from The National Heart, Lung, and Blood Institute (NHLBI, Bethesda, MD, USA) DNA Sequencing and Genomics Core. 

#### 3.1.3. Data Analysis

Reads were initially trimmed for quality and adapter removal using fastp v0.23.2 [12]. Trimmed reads were then analyzed for microbial and viral reads using Kraken2 v2.1.2 [13] and the precompiled Standard database (https://benlangmead.github.io/aws-indexes/k2 (accessed on 1 December 2022)). Kraken2 reports were combined and analyzed using Pavian v1.2.0 [14].

## 4. Expected Results

Current *Bacteroides* heparinase I treatment is well-suited for a wide range of cfRNA concentrations.

In this study, the average concentration of extracted cfRNA was 78.09 ± 10.66 ng/mL of heparin plasma with a range of 10.59–164.04 ng/mL of heparin plasma. The average amount of RNA used for *Bacteroides* heparinase I treatment was approximately 52 ± 7.11 ng (range is 7.06–111.76 ng) (Figure 1).

*Bacteroides* heparinase-I-treated libraries were suitable for metagenomic analysis. 

The High Sensitivity D5000 ScreenTape^®^ Report showed that the average RNA library size was 259 ± 7.21 bp, ranging from 194 to 302 bp (Figure 2C). The classification report from Pavian R package v1.2.0 showed no artificial reads in the samples. Average classified reads were 98.85 ± 0.0006%, and the rest were unclassified (1.05 ± 0.006%). Similar to previous studies [7,15], most of the classified reads belonged to chordates (~59 ± 0.025%), and the rest were microbial reads (~41 ± 0.024%) (Figure 2D). 

## 5. Discussion

Plasma cfRNAs are considered non-invasive markers in different diseases and a potential source for both human and microbial signatures in blood [16,17,18]. Many previous studies analyzing cfRNAs used EDTA plasma, as heparin is a known inhibitor of qPCR-based downstream analysis. As a result, blood is usually collected into various types of tubes, increasing the risk of sampling error, contamination, burden of work, and cost. In the present study, we have designed a cost-effective and straightforward protocol for heparinase treatment (using *Bacteroides* heparinase I enzyme) of plasma stored in sodium heparin for many years at −80 °C followed by next-generation sequencing (NGS) to show feasibility for qPCR-based downstream analysis.

Instability and low concentrations are two significant challenges for the study of heparinized plasma cfRNAs, making it harder to incubate samples for a long time with heparinase enzymes [2]. Plasma cfRNA concentrations depend on the plasma volume, type of subjects, and the extraction procedures. As reported by other studies, a 1 h incubation with enzyme mixtures [8,19] helps in the removal of heparin and also allows successful library preparation, even with meager quantities of starting cfRNAs (~7 ng). Use of the buffer from the same cDNA synthesis and library preparation kit makes this method of heparinase treatment far more time- and cost-efficient than others [8,19]. Furthermore, many of the previously published heparinase treatment studies were performed with fresh plasma, whereas the described approach is suitable for frozen plasma stored for at least up to 6 years at −80 °C.

Though there are differences in the read count and reproducibility in NGS data, it has been well established that a far greater number of plasma cfRNAs align with the human genome compared to microbial genomes [7]. This finding is consistent with the present study. It has been difficult to find studies that deal with plasma stored in sodium heparin, which were then utilized for NGS to analyze both human and microbial sequences, making the described protocol a novel approach for heparinase treatment and subsequent NGS.

There are a few limitations of the current study. To minimize the degradation of cfRNAs, no experiments were performed to investigate the heparin concentration in the samples before and after heparinase treatment. No comparison was performed between EDTA plasma, non-treated heparin plasma, and treated heparin plasma in the present study in terms of extracted cfRNA concentrations and NGS. As a result, the effect of different anticoagulants on plasma cfRNA populations is not addressed in this manuscript. Multiple library samples stored in EDTA and heparin (treated and non-treated with heparinase) plasma from healthy blood donors were analyzed through the 2100 expert_High Sensitivity DNA Assay kit to confirm successful library preparation (Figure 3) after heparinase treatment. As our goal is to establish an authentic approach for heparinase treatment only, we do not present any analysis of microbial species in this paper. Clinical and epidemiological studies can continue over the span of several years, with multiple time points and interventions. It would be efficient and effective if there were one suitable anticoagulant for preserving plasma to perform metabolomics and transcriptomics. Here we present a simple and authentic approach to removing heparin from frozen, heparinized plasma stored for several years and subsequent use for qPCR-based downstream analysis. 

## 6. Conclusions

The current study is a novel and practical approach to heparinase treatment for human plasma collected and frozen in sodium heparin for several years. The protocol is an effective demonstration of the removal of heparin in plasma as a barrier to transcriptomic research.

## Figures and Tables

**Figure 1 mps-06-00112-f001:**
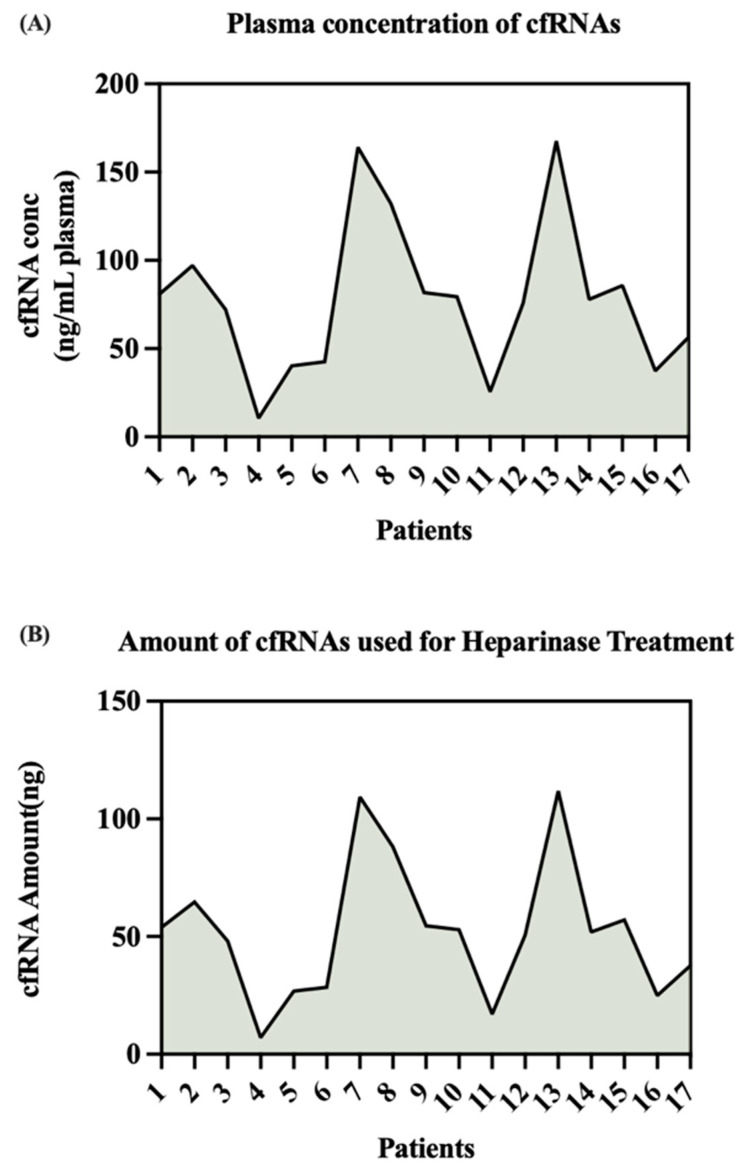
cfRNA concentrations in the HCV heparin plasma. cfRNA concentrations varied from sample to sample (**A**). As 4 μL of water-suspended cfRNAs was directly used (without correction of concentration) for heparinase treatment, starting cfRNA amounts were different for these samples (**B**).

**Figure 2 mps-06-00112-f002:**
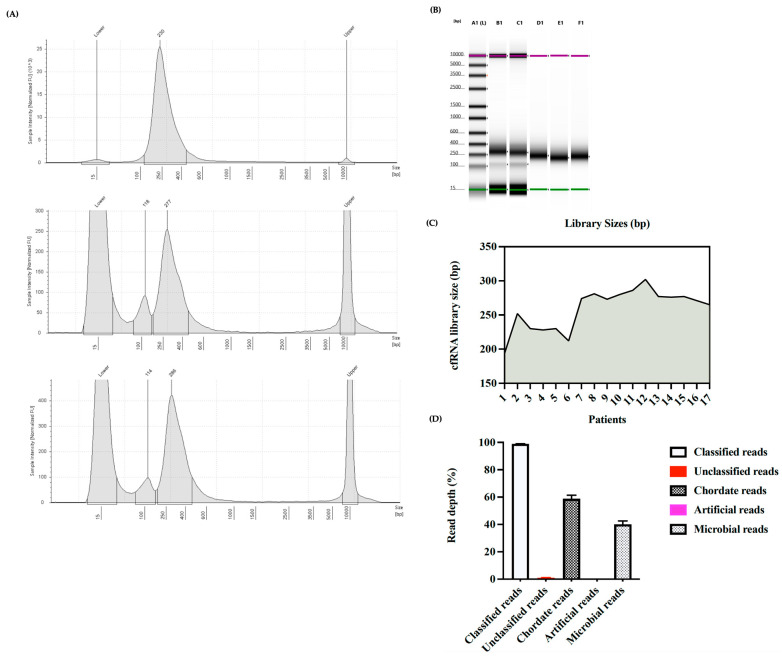
cfRNA libraries from heparinized plasma of HCV patients. Library quality control report from High Sensitivity D5000 ScreenTape^®^ (**A**,**B**). Green and Purple color lines in figure B indicate lowest and highest ladder respectively present during TapeStation analysis. Library sizes for 17 HCV patients (**C**). Percentages of multiple subtypes of reads found in the HCV patient samples (**D**).

**Figure 3 mps-06-00112-f003:**
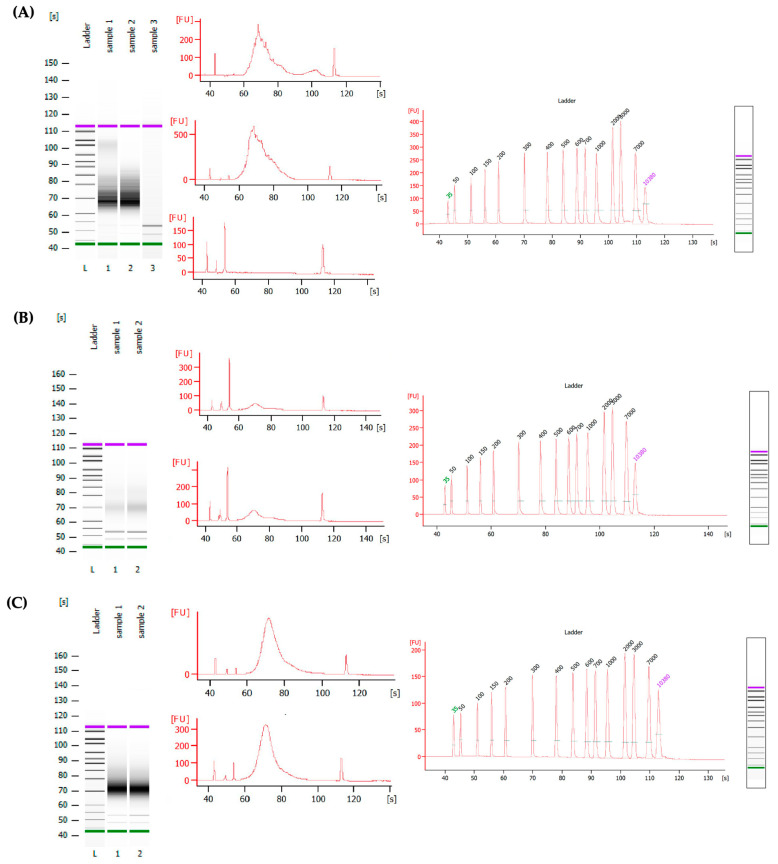
cfRNA libraries from healthy blood donors. Panel (**A**) and (**B**) represent library from *Bacteroides* heparinase-I-treated and non-treated heparin plasma, respectively, and (**C**) represents libraries from EDTA plasma. Experiments were carried out in multiple batches in multiple days. A cfRNA amount of ~8–50 ng per 1 mL of plasma was used for the upper experiments. Green and Purple color lines indicate lowest and highest ladder respectively present in bioanalyzer.

**Table 1 mps-06-00112-t001:** Experimental conditions for heparinase treatment.

	Concentrations or Units	Volume Added (μL)	Final Volume per Sample (μL)
Sample	1.7–27.3 ng/mL plasma	4	11.8
*Bacteroides* Heparinase I	12,000 Units/mL	1.5	
Heparinase Buffer	10X	1.2	
RNAse inhibitor	20 Units/mL	0.5	
NEBNext First Strand Synthesis Reaction Buffer	5X	4.6	

**Table 2 mps-06-00112-t002:** Conditions for PCR enrichment during library preparation.

Step	Temperature	Time	Cycles
Initial Denaturation	98 °C	30 s	1
Denaturation	98 °C	10 s	14
Annealing/Extension	65 °C	75 s	
Final Extension	65 °C	5 min	1
Hold	4 °C	10 min	

## Data Availability

Any data not published in this manuscript is available upon reasonable request from the corresponding author.

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
