# Peer review of "A Simple, Rapid, and Effective Heparinase Protocol to Enable Nucleic Acid Study from Frozen Heparinized Plasma"

_mps, 2023, doi:10.3390/mps6060112_

Round 1
Reviewer 1 Report
Comments and Suggestions for Authors
Interesting new method has been described in this article.
General comment:
It’s a protocol paper so please describe in details everything not briefly or using previous paper.
For example, line 74: cfRNAs were extracted from 1 mL plasma using a previously published method [7]. In brief,
Please detailed cDNA synthesis procedure.
Some points to modify:
In human plasma part: please indicate the volume collected for each tube.
In extraction of RNA part: Is the protocol achievable with less than 1mL. What is the max and min volume usable? => indicate in the manuscript
When the UltraPure Glycogen is used? in same time than the 2-Propanol?
Line 91: Samples treated with Bacteroides Heparinase I were directly taken for cDNA synthesis. Directly means after the 1h at 37°C?
What readouts do you use to consider a sample not to be of good quality?
Why did you choose the specific enzyme “Bacteroides Heparinase I enzyme”?
Your protocol works on frozen plasma. Is it work on fresh plasma too?
Author Response
Thank you so much for the comments and suggestions. Please see the attachment

Reviewer 2 Report
Comments and Suggestions for Authors
This manuscript is aimed to explore the usage of frozen Heparinized plasma for deep sequencing cfRNA. After the treatment of Heparinase, the extracted cfRNA was quantified for this usage by evaluation of the amount of RNA, RNA library size and classified reads. It's a simple way to deal with the extracted RNA from Heparinized plasma, and gives valuable suggestions for the usage of clinical samples. Moreover, the manuscript is well written in general.
This manuscript is meanful for stored valueable samples. I don't think it can be used in routine practice as lots of precesses needed for NGS or other nucleic acid related experiments, unless the comparison experiments have be done on samples with various anticoagulation.
Author Response

(The authors gave the same response as above.)
